# Age at menarche and its association with preschool BMI among girls in Northern Norway

Henrik Lykke Joakimsen[1‡]*, Astrid Brendlien[1‡], Anne-Sofie Furberg[2,3], Christopher Sivert Nielsen[4,5], Guri Grimnes[1,6], Elin Kristin Evensen[7]

1 Tromsø Endocrine Research Group, Department of Clinical Medicine, UiT The Arctic University of Norway, Tromsø, Norway, 2 Department of Microbiology and Infection Control, University Hospital of North Norway, Tromsø, Norway, 3 Faculty of Health and Social Sciences, Molde University College, Molde, Norway, 4 Department of Chronic Diseases, Norwegian Institute of Public Health, Oslo, Norway, 5 Department of Pain Management and Research, Oslo University Hospital, Oslo, Norway, 6 Division of Internal Medicine, University Hospital of North Norway, Tromsø, Norway, 7 Department of Health and Care Sciences, Faculty of Health Sciences, UiT the Arctic University of Norway, Tromsø, Norway.

‡ These share first-authorship.
* henrik.l.joakimsen@uit.no

## Abstract

### Background

A decreasing age of menarche has been reported across the Western world. Early menarche is associated with unfavorable health outcomes.

### Aim

The aims of this study were to describe the age at menarche in a general population sample in Norway and the associations between body mass index (BMI) categories at preschool (approximately 6 years of age) and age at menarche.

### Methods

We used self-reported age at menarche among girls who participated in the population-based study Fit Futures 1 (FF 2010–2011), mostly born in 1994, to calculate age at menarche. The preschool BMI from health records was divided into BMI categories according to validated cutoffs on the basis of age and sex from the International Obesity Task Force (IOTF). We estimated the effect of preschool BMI on age at menarche via a linear regression model adjusted for socioeconomic status (SES).

### Results

Among 500 girls with a mean age of 16.5 years (standard deviation (SD) ± 1.4), 497 (99%) had completed menarche. The mean age at menarche was 13.0 years (SD ± 1.2). According to the fitted linear regression model, preschool obesity was a statistically significant predictor of age at menarche and was associated with

**Data availability statement:** The data supporting the findings of this study are provided by The Fit Futures study (Third-party). However, access to these data are restricted due to local and national ethical and security policies. Data are available from The Fit Futures study on reasonable request, following the steps presented on their website (https://uit.no/research/fitfutures_en#region_837245). Researchers can replicate our study findings in their entirety by directly obtaining the data from the Fit Futures (part 1) and following the information outlined in our Methods section. In our case, some of the authors were involved in collecting data for Fit Futures, which may have improved the speed of accessing the data compared to external researchers that requests the same. Requests for data access should be directed to The Fit Futures study at UiT, The Arctic University of Norway, Department of Community Medicine, via email (fitfutures@uit.no).

**Funding:** The author(s) received no specific funding for this work.

**Competing interests:** The authors have declared that no competing interests exist.

menarche 9.5 months earlier than a normal preschool BMI was. $R^2$ estimated that preschool BMI could explain 3% of the variance in age at menarche.

## Conclusion

The mean age at menarche in Northern Norway was 13.0 (SD ± 1.2) years, similar to previous Norwegian studies. Childhood obesity was associated with earlier age at menarche.

---

## 1 Introduction

Menarche, the first menstruation, occurs at a younger age than before across the Western world. This trend seems to be closely related to increasing living standards [1]. From an evolutionary perspective, earlier menarche means an earlier opportunity for reproduction, ensuring species survival. Early menarche is, however, associated with several unfavorable outcomes, such as increased risk of breast cancer, depression, chronic pain, type 2 diabetes, cardiovascular disease, and overall mortality [2–4]. Age at menarche has decreased with increasing standards of living in the past. In Norway, the mean menarche age fell from 15.6 to 14.6 years between 1860 and 1880, during a period of great improvement in welfare. The mean age at menarche decreased again soon after the second world war to approximately 13.3 years for women born after 1940 [5]. Recent studies have shown another sudden decrease in age at menarche across several western countries, such as the Netherlands, Canada, the United States, and Norway, during the last 20 years [2,6,7]. A recent Norwegian study published in 2020 reported a decrease from a mean age of 13.3 to 13.1 years from 2003–2006 to2016 [2]. This is similar to the mean age at menarche of 13.2 (SD ± 1.3) years for nulliparous women and 13.1 (SD ± 1.4) years for parous women found among healthy women aged 25–35 years who participated in the Energy Balance and Breast Cancer study (EBBA-1) in our region (Tromsø and surrounding municipalities, North Norway) from 2000–2001[8]. Menarche timing is known to vary across ethnicities[9] and socioeconomic statuses (SESs)[10]. Increasing living standards and nutritional status suggest that obesity, or overnutrition, can be a cause of the trend of earlier menarche in developed countries. Obesity is associated with early menarche through several pathways. High leptin levels from adipose tissue may signal the body to initiate puberty, and high aromatase activity can enhance peripheral conversion of androgens to estrogens [11]. Additionally, obesity-related hyperinsulinemia can stimulate release of GnRH from the hypothalamus and reduce sex hormone-binding globulin levels, making more sex hormones available, which also can trigger earlier pubertal development[12]. The prevalence of obesity is increasing globally in both children and adults [13]. Approximately one in six children in Norway are overweight or obese [14], with the highest prevalence in the rural north [15]. Several studies have explored the associations between early menarche and obesity in adolescence or adulthood [12]. In our study, however, we wanted to investigate the effect of childhood obesity on age at menarche, for which

the literature is limited. We identified a systematic review that showed that since 2017, six studies have reported a significant association between high BMI in childhood and early menarche[11]. Questions of whether there is a change in age at menarche, if the change is local, still in motion, and causes, remain to be answered. We hypothesized that the age at menarche is lower in Northern Norway compared to other Norwegian populations, and that there is an association between higher BMI in childhood and early menarche.

The primary aims of this study were therefore to describe age at menarche in a general population sample in Northern Norway, and the association between body mass index (BMI) at preschool age and age at menarche.

## 2 Materials and methods

### 2.1 Study population

The Fit Futures (FF) study is a population-based longitudinal study of lifestyle and health among adolescents in Northern Norway. In the period 20.09.2010–27.04.2011, all first-year upper-secondary students (mainly aged 15–16 years) from all schools in the Tromsø and Balsfjord municipalities were invited to FF (n = 1117), and 1038 (93%) participated. Among the 508 girls included in the study, the majority were born in 1994 with birth years ranging from 1982–1994. This cohort included some older girls and women (n = 79) who attended upper-secondary school later than most students, due to various reasons such as previous school dropout or immigration. The students met at the Clinical Research Unit of The University Hospital of Northern Norway, where they completed a digital questionnaire and blood sampling, clinical examinations and interviews performed by study nurses. Data were accessed for research purposes 14.06.2021. Further details of FF1 are described elsewhere[16].

### 2.2 Variables

Age at menarche in years and months and parental education were assessed via a digital self-administered questionnaire. Girls were surveyed with the following questions: "Have you experienced menstruation? (yes/no)" and "If yes (*have you experienced menstruation*), how old were you when you experienced menstruation the first time? ([Age in whole years] [months])". Participants who reported age at menarche in years but did not specify months (n = 179) were assumed to be 0 months beyond years at menarche. The robustness of this assumption was tested by repeating all analyses after changing it to 6 months beyond years at menarche. Parent education was surveyed regarding both parents as the highest education completed according to five categories: "Primary school", "Vocational training", "Upper-secondary", "Higher education < 4 years", or "Higher education ≥ 4 years". Parent education was used as a proxy for SES. SES was classified as high or low, where low SES was defined as having parents with no further education after upper-secondary school, and high SES was defined as having one or both parents who had education beyond upper-secondary school. The study population consisted of 97% Caucasian participants, so ethnicity was discarded as a relevant predictor. Further details of the questionnaire used in FF can be found on the homepage [17].

Height and weight at preschool routine check-up, at approximately 6 years of age, were gathered retrospectively between 2015 and 2018 from physical health records stored in archives of the municipality administrations. Data on height, weight and age in years and months at the date of measurement, were transferred to the FF database. BMI was calculated as weight in kilograms divided by height in meters squared. BMI categories were defined by age- and sex-specific cutoffs suggested by the International Obesity Taskforce (IOTF) for children 2–18 years [18]. Participants exact age was merged into half-year age intervals and grouped by BMI-categories according to that half-year age specific cut-off value for girls. Cut-offs corresponding to an adult BMI < 18.5 kg/m² were defined as underweight, BMI > 18.5 and < 25 kg/m² as normal weight, BMI ≥ 25 and < 30 kg/m² as overweight, and BMI ≥ 30 kg/m² as obese.

## 2.3 Statistical analyses

Eight girls did not respond on any questions related to menarche, leaving 500 girls for the present analysis. The normal distribution of age at menarche was assessed via visual inspection of residual plots. Descriptive statistics, including means with standard deviations (SDs) and ranges, were used to present age at menarche. Correlations between independent (dichotomized) variables were assessed by the Phi coefficient, where a Phi < 0.10 was deemed acceptable. A linear regression model was fitted to estimate the effect of preschool BMI on age at menarche. The model was adjusted for SES. As all the girls in upper-secondary schools within the targeted study population were invited to participate, selection bias was considered low, and no a priori power analyses or randomization were conducted. In the present study, all available data for the girls are included.

Girls who had not reached menarche (n = 3) and participants who had responded to whether they had experienced menarche but did not specify their age (n = 8) were defined as having an age at menarche equal to the age at inclusion in FF. This was done to reduce estimate bias toward 0 and was deemed fitting owing to the low number affected in the data (1%).

Girls with missing preschool BMI data (n = 124) were not included in the regression model but were included in the descriptive statistics of age at menarche.

Statistical analyses were performed in R version 4.1.2 (R Foundation for Statistical Computing, Vienna, Austria, 2021].

## 2.4 Ethics approval and consent to participate

All participants in FF signed an informed consent form. Those younger than 16 years had to provide additional written consent from a guardian. The study was conducted in accordance with the Declaration of Helsinki. The current study was approved by the Regional committee for medical and health research ethics north (reference number 213685/REK nord) in January 2021.

## 3 Results

The mean age of the 500 girls was 16.4 (SD ± 1.3, range 15–28) years. Four hundred ninety-seven (99%) of the 500 included girls had completed menarche (Fig 1). Age at menarche was normally distributed, with a mean of 13.0 years (SD ± 1.2), and there was a trend toward decreasing age at menarche with increasing BMI (Fig 2). Age at menarche did not differ by SES. There was low correlation between BMI and SES (the strongest was phi coefficient 0.044 between *low SES* and *underweight in preschool*, p = 0.55). Among 374 girls with available preschool health records, the mean age at preschool examination was 6.1 years, which was normally distributed, with a range of 4.0–8.2 years.

The mean and median age at menarche by BMI category at preschool age and SES are shown in Table 1, which also shows the trend of decreasing age at menarche by BMI category.

The overall regression was statistically significant ($R^2$ = 0.03, p = 0.03), and is shown in table 2. There seemed to be an inverse linear relationship between increasing preschool BMI and age at menarche, but only preschool obesity was significantly associated with age at menarche, which was estimated to be 0.80 years (9 months and 18 days) earlier (p < 0.01) than that of preschool normal weight participants.

### 3.1 Sensitivity analyses

All the assumptions of all the analyses were tested and met.

When participants with missing month data for age at menarche (n = 179) were recorded from "missing month = 0 months" to "missing month = 6 months", the regression estimates did not change significantly (with alfa = 0.05) (from a mean of 13.0 to 13.1).

Compared with those who had available health records, participants without available health records from preschool (n = 124) were not significantly different (p < 0.05) in terms of age at menarche (Fig 2) or distribution of SES.

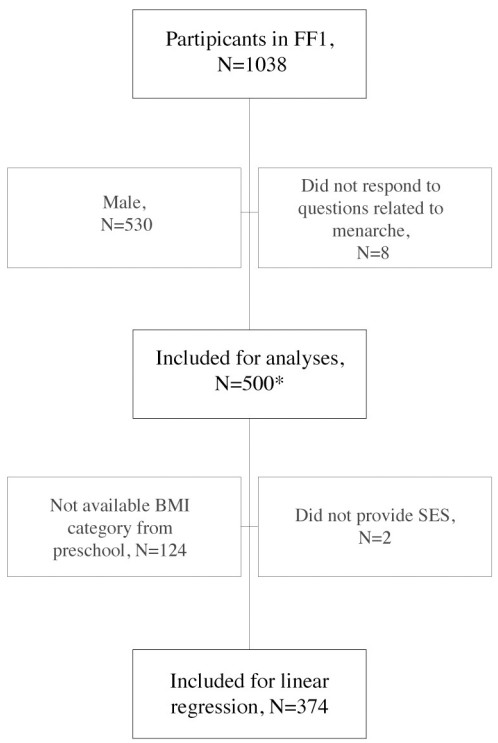

**Fig 1. Participant flow chart.** *Three (3) of the included participants had not reached menarche (aged 16, 16 and 17 years), and eight (8) had reached menarche but did not specify at what age and were defined as having an age at menarche equal to the age at inclusion.

## 4  Discussion

In this study, we found that the mean age at menarche in girls in Northern Norway was 13.0 years. Children with pre-school obesity experienced menarche approximately nine months earlier than normal weight children did, whereas other BMI categories had no significant ability to predict age at menarche. Age at menarche was not significantly different across SESs in multivariate analyses. Our findings of age at menarche are consistent with other studies in Scandinavia[19,20], even though our study population was very homogenous (97% Norwegian) and more obese than those in comparable studies. Our study adds to the understanding of trends in the age at menarche.

We found a lower mean age at menarche than did similar studies from other regions of Norway from the last two decades [2,19]. However, the Bergen Growth Study (BGS), which reported a significant decrease in the mean age at menarche from 2003–2016 from 13.3 to 13.1 years, also noted that approximately 70% of participants had not yet reached menarche [2], making comparisons ambiguous. In the Young-HUNT study with data from 2000–2001[20], the mean age at menarche was 13.2 (12.0–14.4, 95% CI) years, and 78% of the study population had completed menarche [19]. Our confidence intervals overlap with those of the Young-HUNT study, thus not indicating differences across our populations [19]. The same was the case with the study of Iversen et al., who also investigated age at menarche in the same region as our study (Tromsø area) in 2000–2001, and reported a mean age at menarche of 13.2 (SD ± 1.32) years for nulliparous women and 13.1 (SD ± 1.42) years for parous women [8]. Accordingly, we cannot conclude from our results that age at menarche has continued to decrease in Norway over the last two decades.

To the best of our knowledge, no other Norwegian study has shown associations between measured preschool BMI and age at menarche. Our findings are consistent with those of a larger Danish study from 2020, which concluded that

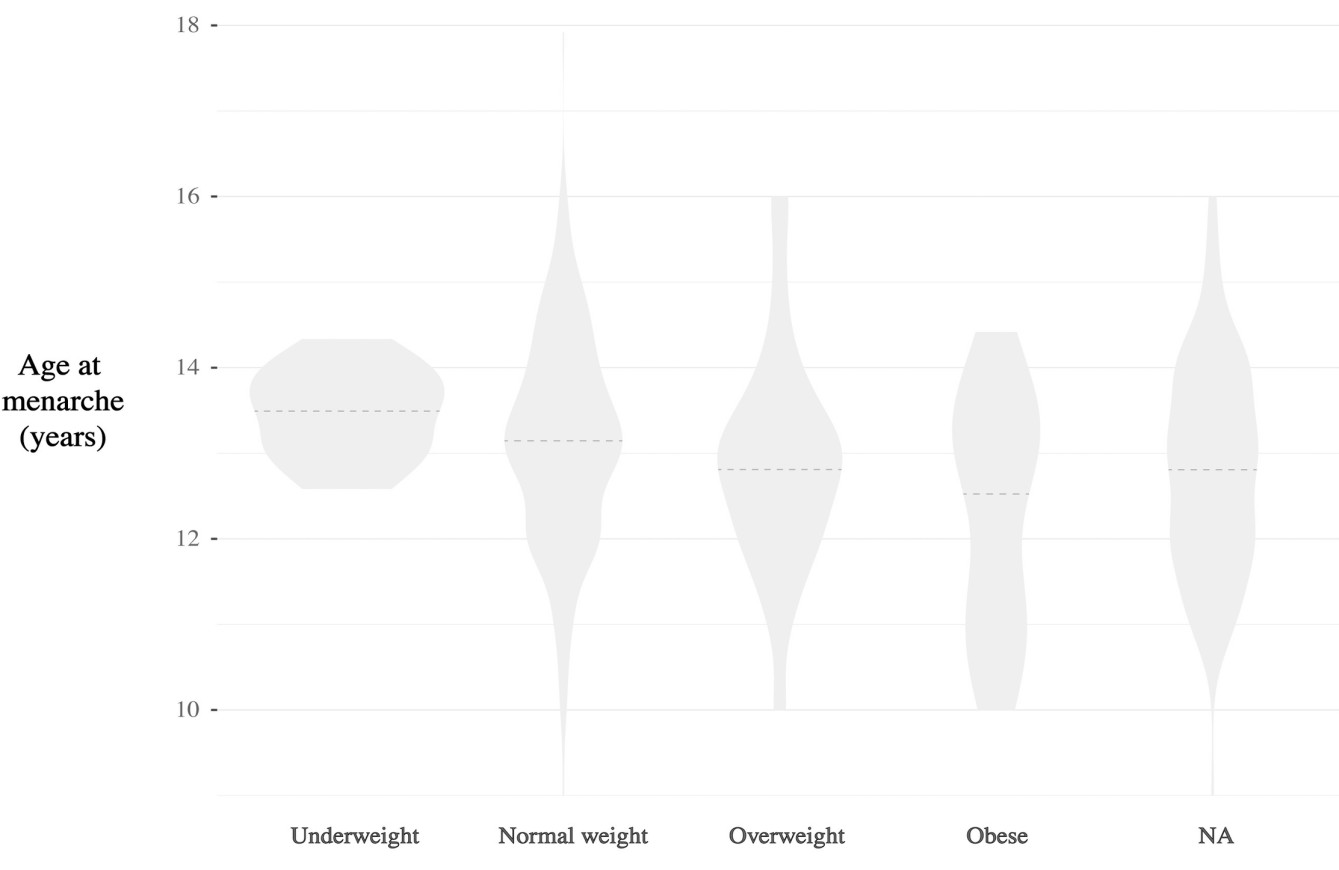

**Fig 2. Violin plot: Age at menarche according to BMI categories at preschool age (n = 374).** Violin plot showing the distribution of participants' age at menarche according to their BMI categories at preschool (approximately 6 years). NA are participants without available preschool records of BMI. The dotted line in each violin represents the mean age for each group. The Fit futures study, N = 500 (NA=124).

**Table 1. Age at menarche across BMI categories at preschool age and SES. Fit Futures study, n = 500.**

| Group (n,%) | Mean age at menarche in years (SD) | Range for age at menarche in years |
|---|---|---|
| **All (n = 500)**[*] | 13.0 (1.2) | 9.0-17.9 |
| **BMI category in preschool** | | |
| Underweight (n = 15, 3%) | 13.5 (0.5) | 12.6-14.3 |
| Normal weight (n = 263, 53%) | 13.2 (1.2) | 9.0-17.9 |
| Overweight (n = 81, 16%) | 12.9 (1.3) | 10.0-16.0 |
| Obese (n = 17, 3%) | 12.3 (1.4) | 10.0-14.4 |
| Missing (n = 124, 25%) | 12.8 (1.2) | 9.0-16.0 |
| **Socioeconomic status** | | |
| High (n = 247, 49%) | 13.0 (1.2) | 9.0-16.0 |
| Low (n = 251, 50%) | 13.0 (1.3) | 9.0-17.9 |
| Missing (n = 2, < 1%) | 12.3 (0.4) | 12.0-12.6 |

Mean age at menarche and range.

[*]Three (3) of the included participants had not reached menarche (aged 16, 16 and 17 years), and eight (8) had reached menarche but did not specify at what age and were imputed to have age at menarche equal to age at inclusion.

**Table 2. Linear regression model - Age at menarche across BMI category at preschool age and SES. Fit futures study, n = 500.**

| Predictors | Age at menarche | | |
|---|---|---|---|
| | Estimates | CI | p |
| (Intercept) | 13.19 | 13.00–13.39 | **<0.001** |
| Underweight | 0.32 | -0.32–0.96 | 0.322 |
| Overweight | -0.28 | -0.59–-0.03 | 0.076 |
| Obese | -0.80 | -1.41 – -0.20 | **0.009** |
| Low SES | -0.08 | -0.33–0.17 | 0.509 |
| Observations | 374 | | |
| $R^2$ /$R^2$ adjusted | 0.029/0.019 | | |

$\beta_0$ represents the estimated age at menarche in girls in the normal weight BMI category in preschool and of high socioeconomic status (SES), and $\beta_{Underweight}$, $\beta_{Overweith}$, $\beta_{Obese}$, $\beta_{Low\ SES}$ represents dummy variables.

a higher BMI around seven years of age was associated with younger age at attaining all pubertal milestones, including menarche, in a dose-dependent manner, irrespective of socioeconomic factors [21]. Furthermore, a systematic review from 2017 on longitudinal studies on birth weight, early weight gain and age at menarche also revealed an association between high childhood weight and lower age at menarche. They also reported an association between rapid weight gain in childhood and early menarche [11].

SES was not associated with age at menarche in our population. This finding was in accordance with findings from the BGS [2]. This may be due to the low degree of social inequality in Norway [22], where even those with low SES have sufficient nutrition available to reach menarche.

This study has several strengths, including the use of a population-based sample and a high participation rate in which nearly all participants had completed menarche. The risk of memory bias was low, as the survey was carried out few years after participants had experienced menarche, as was the fact that most women have good recollection of age at menarche even after many years [23]. BMI is a well-established and useful measure to estimate the amount of body fat in children and adolescents [24]. BMI should, however, not be confused as a precise measure of body fat [24]. Preschool height and weight were measured by health personnel, which are preferable to self-reports [25].

This study also has several limitations. The IOTF reference values, i.e., the age- and sex-stratified BMI cutoffs, are known to have low sensitivity and high specificity, which might lead to a falsely low estimated prevalence of overweight and obesity [18]. Furthermore, in this study, we only had a single measurement of preschool BMI, which can vary during childhood [16]. Parent education level, as a measure of socioeconomic status, is known to be imprecise, even though it is commonly used [2]. Even though age at menarche is known to vary by ethnicity[9], our sample was not fit to describe such variations because our sample consisted of 97% Caucasians. On one hand, that makes controlling and interpretation easier, while also making our findings less applicable to other populations. Age at menarche is known to be heritable [20,26,27], but genetic profile and the mother's age at menarche was not available in our study. Other factors that may impact the timing of menarche are genetic factors, endocrine-disrupting chemicals, maternal smoking during pregnancy and low birth weight [27–29].

## 5 Conclusion

The mean age at menarche in northern Norway was 13.0 (SD±1.2) years, consistent with previous Norwegian studies. While childhood obesity was associated with earlier age at menarche, it accounted for only a small part of the variation. This suggests that other factors may also play substantial roles. Future research should explore additional determinants, including estimation of effect size, and assess effectiveness of targeting childhood obesity. Our findings support early

preventive efforts against obesity; however, clinicians aiming to prevent early menarche should adopt a broader approach rather than focusing solely on obesity.

## Supporting information

**S1 File. Supporting information – Declarations**
(DOCX)

## Author contributions

**Conceptualization:** Henrik Lykke Joakimsen, Astrid Brendlien, Anne-Sofie Furberg, Christopher Sivert Nielsen, Guri Grimnes, Elin Kristin Evensen.

**Data curation:** Anne-Sofie Furberg, Christopher Sivert Nielsen, Elin Kristin Evensen.

**Formal analysis:** Henrik Lykke Joakimsen, Astrid Brendlien, Guri Grimnes.

**Investigation:** Elin Kristin Evensen.

**Methodology:** Henrik Lykke Joakimsen, Anne-Sofie Furberg, Christopher Sivert Nielsen, Guri Grimnes, Elin Kristin Evensen.

**Project administration:** Guri Grimnes.

**Supervision:** Guri Grimnes, Elin Kristin Evensen.

**Visualization:** Henrik Lykke Joakimsen.

**Writing – original draft:** Henrik Lykke Joakimsen, Astrid Brendlien.

**Writing – review & editing:** Henrik Lykke Joakimsen, Astrid Brendlien, Anne-Sofie Furberg, Christopher Sivert Nielsen, Guri Grimnes, Elin Kristin Evensen.

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
