## [Decision Letter · Decision Letter 0]

13 Feb 2025

PONE-D-25-00972Age at menarche and its association with preschool BMI among girls in Northern NorwayPLOS ONE

Dear Dr. Joakimsen,

Thank you for submitting your manuscript to PLOS ONE. After careful consideration, we feel that it has merit but does not fully meet PLOS ONE’s publication criteria as it currently stands. Therefore, we invite you to submit a revised version of the manuscript that addresses the points raised during the review process. Please mind that the reviewer's remark about the absence of novelty is not a criterium for publication in PLOS :)

We look forward to receiving your revised manuscript.

Kind regards,

Inge Roggen, M.D., Ph.D.

Academic Editor

PLOS ONE

Journal Requirements:

Reviewers' comments:

Reviewer's Responses to Questions

**Comments to the Author**

1. Is the manuscript technically sound, and do the data support the conclusions?

Reviewer #1: Yes

Reviewer #2: Yes

2. Has the statistical analysis been performed appropriately and rigorously? 

Reviewer #1: Yes

Reviewer #2: Yes

3. Have the authors made all data underlying the findings in their manuscript fully available?

Reviewer #1: Yes

Reviewer #2: Yes

4. Is the manuscript presented in an intelligible fashion and written in standard English?

Reviewer #1: Yes

Reviewer #2: Yes

5. Review Comments to the Author

Reviewer #1: Joakimsen, et al. report data from a population-based study of growth and development in girls from Norway. They identified 500 girls who were in secondary school and who had self-reported information available on the timing of menarche. Most of those girls also had preschool records of height and weight. These data were used to determine the average age of menarche and to assess relationships between BMI category and the timing of menarche. The study’s strength is the large number of overall participants and the population-based nature of the data. However, the number of underweight and obese girls was very low, and the population was very homogenous, limiting the broader applicability of the data. It is not clear what this paper adds to the literature, as there are a number of other papers showing the same findings.

Specific comments:

1. The population studied is not clear. The authors included girls who were in the first year of upper secondary school in 2010-2011, but they also state that they were born between 1982-1994. They also state that the age range was 15-28 years. Clarification is needed.

2. Participants were asked to indicate their age at menarche in years + months, but some reported only years, requiring the authors to make assumptions about the exact age. How many girls did not report years + months?

3. Participants’ BMIs were categorized as underweight, normal, overweight, and obese based on criteria established in adults. It is not clear how these cutoffs were applied to their data. Was this done by determining the corresponding percentiles for a pediatric population?

4. In the Study Population section, near the end of the paragraph, the text should read “…clinical examinations and interviews performed by study…”

5. In the Statistical Analysis section, 2nd paragraph, the authors state that 8 girls did not specify their age at menarche and had an age assigned. However, in the preceding paragraph, they state that these 8 girls were excluded. Clarification is needed.

6. The age at menarche did not differ across the two SES categories. I wonder if there was any relationship between SES and BMI status.

7. In the Discussion section, 5th paragraph, the sentence “BMI is a well-established and useful measure to estimate the amount of body fat that children and adolescents consume” does not make sense and should be reworded.

8. The information in the legend to Table 1 also appears in the text and can be deleted.

9. As mentioned, it is not clear what this paper adds to the body of knowledge, as their findings have been demonstrated in other studies, including in Scandinavian populations.

Reviewer #2: The article focuses on an important issue, the onset of menarche and the association with obesity in Norway. The study is classically constructed, the topics are relevant, and the design is clear. A number of questions arose during the review:

1. The introduction should include information on how obesity is associated with early menarche, what are the pathways?

2. no hypothesis of the study

3. the formula and full calculation of the study sample should be stated, and how randomization was done, since this is an epidemiological study

4. the conclusion lacks information about further ways of work, how to use this information in practice, etc.

5. in the list of references there are sources older than 5 years where it is possible to replace them

6. PLOS authors have the option to publish the peer review history of their article (what does this mean? ). If published, this will include your full peer review and any attached files.

**Do you want your identity to be public for this peer review?** For information about this choice, including consent withdrawal, please see our Privacy Policy .

Reviewer #1: No

Reviewer #2: **Yes: ** Pofessor Mariia V. Matveeva

---

## [Author Response · Author response to Decision Letter 1]

25 Mar 2025

Dear editor and reviewers,

Thank you for the opportunity to revise our manuscript and for your constructive and helpful suggestions to improve it. We have carefully read and amended our manuscript according to your comments. Firstly, we have made sure that everything is formatted correctly according to PLOS ONEs style requirements and submission guidelines, updated the data availability statement in the supporting information, and included the ethics statement in the methods section. Further, we have responded to each specific comment below in the same ordered as we received them. Changes according to reviewer comments are highlighted in yellow in the file “Revised manuscript with Track Changes”.

Reviewer #1

(reviewer comment in normal font, response in cursive):

1. The population studied is not clear. The authors included girls who were in the first year of upper secondary school in 2010-2011, but they also state that they were born between 1982-1994. They also state that the age range was 15-28 years. Clarification is needed.

The text is revised to more clearly explain how these statements are not in conflict with each other – rather it shows that we had some older girls (n=79) and women who attended upper-secondary together with the 15-16-year-olds. See 3.1 Study population, sentence no.3 and 4.

2. Participants were asked to indicate their age at menarche in years + months, but some reported only years, requiring the authors to make assumptions about the exact age. How many girls did not report years + months?

The number of girls who did not report years + months is added in the text (n=179). See 3.2 Variables, sentence no.3.

3. Participants’ BMIs were categorized as underweight, normal, overweight, and obese based on criteria established in adults. It is not clear how these cutoffs were applied to their data. Was this done by determining the corresponding percentiles for a pediatric population?

A sentence explaining the BMI-cut-off coding has been added in the manuscript. See 3.2 Variables, second paragraph, sentence no.4 and 5.

If it is still unclear: For BMI categorization, we followed the IOTF pediatric age- and sex-adjusted BMI, which consists of a vast table with different adjustements for every 6 months of age for each gender. We coded each participant (all girls) into category of age, i.e., for preschool BMI we adjusted by age-groups such as “6.0-6.4 years”, “6.5-7.0 years”. Then we coded the age-specific cut-offs for these age-ranges for girls. An example of the specific lines of code looked like this:

Data[Data$BMI_PRESCHOOLCONT_FF1 < 13.385 & Data$agegroups=="6.0-6.4" ,

"WeightClass"] <- "Underweight"

We coded each age interval for each BMI category like this.

4. In the Study Population section, near the end of the paragraph, the text should read “…clinical examinations and interviews performed by study…”

The abundant “were” is removed from the manus. See 3.1 Study population, sentence

no.6.

5. In the Statistical Analysis section, 2nd paragraph, the authors state that 8 girls did not specify their age at menarche and had an age assigned. However, in the preceding paragraph, they state that these 8 girls were excluded. Clarification is needed.

The first sentence has been edited for clarification. See 3.3 Statistical analyses, first paragraph, sentence no 1.

If yet unclear: The girls were asked “Have you experienced your first menstruation?”, as well as “If so, how old were you”. We excluded those 8 who did not respond to any questions of menarche nor menstruation. We included those who had responded to NOT having menarche (n=3), and those who had responded to have menarche but DID NOT specify age (n=8), and imputed them by defining their age of menarche as age of inclusion (which was around 16 years for all n=3+8). This reduced the bias towards 0, which we think gives a picture closer to the real one than if we excluded them all.

6. The age at menarche did not differ across the two SES categories. I wonder if there was any relationship between SES and BMI status.

We found low correlation between SES and BMI – added in the manuscript. See 4 Results, first paragraph, sentence no. 5.

7. In the Discussion section, 5th paragraph, the sentence “BMI is a well-established and useful measure to estimate the amount of body fat that children and adolescents consume” does not make sense and should be reworded.

The phrase has been reworded in the manuscript. See Discussion, fifth paragraph, sentence no.3.

8. The information in the legend to Table 1 also appears in the text and can be deleted.

Abundant parts of the legend in table 1 are removed in the manuscript.

9. As mentioned, it is not clear what this paper adds to the body of knowledge, as their findings have been demonstrated in other studies, including in Scandinavian populations

We have added a sentence about the contribution of the current study. See Discussion, first paragraph, sentence no. 4.

Reviewer #2:

1. The introduction should include information on how obesity is associated with early menarche, what are the pathways?

Context on pathways between obesity and early menarche has been added in the introduction. See Introduction, first paragraph, sentence no13-14.

2. No hypothesis of the study

The hypothesis of the study is added in the manuscript. See Introduction, first paragraph, sentence no.21.

3. The formula and full calculation of the study sample should be stated, and how randomization was done, since this is an epidemiological study

All students from the first year of upper-secondary from all schools in Tromsø and Balsfjord municipalities(the targeted study population) were asked to participate. Initially, the Fit Futures study was considered part of the youth cohort in the population based Tromsø study (T6). As all were invited, and participation were high (93%), we deemed the risk of selection bias low, and did therefore no randomization or power calculations. We have revised the manuscript to describe this. See 3.3 Statistical Analyses, first paragraph, sentence no.7 and 8.

4. The conclusion lacks information about further ways of work, how to use this information in practice, etc.

Information on future ways of work and clinical utility of our findings are added in the conclusion of the manuscript. See 6 Conclusion, sentence no.2-5.

5. In the list of references there are sources older than 5 years where it is possible to replace them

The reviewer comments that some of our sources are older than 5 years and can be replaced. All references older than 5 years are listed and commented below:

• Zhang Z, 2019, Early age at menarche is associated with insulin resistance: a systemic review and meta-analysis

o Removed and replaced by more recent study.

• Krieger, 2014, Age at menarche: 50-year socioeconomic trends among US-born black and white women

o This study has such a long follow-up that we deem it relevant for the overview and understanding of menarche trends over time. We therfore choose to keep this.

• Brudevoll, 1979, Menarcheal age in Oslo during the last 140 years

o As this is the only proper source we found describing menarche in Norway from the 1800s, we want to keep this when giving context on the menarche trends over long time.

• Herman-Giddens ME, 1997, Secondary sexual characteristics and menses in young girls seen in office practice: a study from the Pediatric Research in Office Settings network

o Removed and replaced by a newer study.

• Talma H, 2013, Trends in menarcheal age between 1955 and 2009 in the Netherlands

o As no newer studies from Netherlands was identified, we wanted to keep this source as it gives relevant context. The Dutch and Norwegian population are known to have many similarities.

• Iversen A, 2011, Ovarian hormones and reproductive risk factors for breast cancer in premenopausal women: the Norwegian EBBA-I study

o Only study identified who reported menarche in the same area as our study. We therefore find it proper to keep it for relevant context.

• Meyer H, 2017, Overvekt og fedme i Noreg

o The web page report was written in 2017, but updated in 2023 and remains as such pretty updated. The numbers are still consistent to what we initially reported in the manuscript.

• Kim Y, 2019, Early Menarche and Risk of Metabolic Syndrome: A Systematic Review and Meta-Analysis

o As we mention it mainly to point out that many has studied how early menarche contributes to risk of obesity, it remains a relevant citation and we therefore consider it relevant to keep it.

• Juul, 2017, Birth weight, early life weight gain and age at menarche: a systematic review of longitudinal studies. Obes Rev 2017

o We identified no more recent systematic reviews with the same level of evidence exploring the same aim as ours – rather we only found those who report correlation per grams of protein (in similar groups) or related to infant nutrition. We still deem this source as relevant, and therefore choose to keep it.

• Evensen E, 2017, The relation between birthweight, childhood body mass index, and overweight and obesity in late adolescence: a longitudinal cohort study from Norway:

o We reference further details of Fit Futures, thus it is not relevant to instead cite a more recent study. We want to keep this.

• Cole TJ and Lobstein T, 2012: Extended international (IOTF) body mass index cut-offs for thinness, overweight and obesity

o These cut-offs are still the current recommendatons for the definitions , and therefore kept as is.

• Bratberg GH, 2007, Early sexual maturation, central adiposity and subsequent overweight in late adolescence. a four-year follow-up of 1605 adolescent Norwegian boys and girls: the Young HUNT study

o Represents viable comparison from region within Norway within the same time frame we are discussing in our study (inbetween the last study from our region and us), and we deem it relevant and therefore keep it as is.

• Epland JT, 2019, Slik måler SSB ulikhet

o As our data was collected 2010-2011, data describing social equality closer to time of data collection is more relevant than more recent calculations. We want to keep it as is.

• Simmonds M, 2015, The use of measures of obesity in childhood for predicting obesity and the development of obesity-related diseases in adulthood: a systematic review and meta-analysis

o Removed and replaced by a more recent systematic review. However, not necessary to rephrase wording because the findings were the same.

• Wells JC, 2007, Programming of body composition by early growth and nutrition

o Removed.

• Aksglaede L, 2009, Recent decline in age at breast development: the Copenhagen Puberty Study

o Even though it is not very recent, we deem the study good and relevant to keep in our references. However, we added also a more recent study describing large genetic data associated with menarche.

• Rubin C, 2009, Timing of maturation and predictors of menarche in girls enrolled in a contemporary British cohort

o Same as above (Aksglaede).

• Behie AM and O'Donnell MH, 2015, Prenatal smoking and age at menarche: influence of the prenatal environment on the timing of puberty

o We identidied no better studies that were more recent, and want as such to keep it as in our manuscript.

Other

In the methods-section of the abstract we wrote “self-reported” twice in the first sentence. In the revised version, we have removed one.

---

## [Decision Letter · Decision Letter 1]

1 Apr 2025

Age at menarche and its association with preschool BMI among girls in Northern Norway

PONE-D-25-00972R1

Dear Dr. Joakimsen,

We’re pleased to inform you that your manuscript has been judged scientifically suitable for publication and will be formally accepted for publication once it meets all outstanding technical requirements.

Kind regards,

Inge Roggen, M.D., Ph.D.

Academic Editor

PLOS ONE

Additional Editor Comments (optional):

Reviewers' comments:

Reviewer's Responses to Questions

**Comments to the Author**

1. If the authors have adequately addressed your comments raised in a previous round of review and you feel that this manuscript is now acceptable for publication, you may indicate that here to bypass the “Comments to the Author” section, enter your conflict of interest statement in the “Confidential to Editor” section, and submit your "Accept" recommendation.

Reviewer #1: All comments have been addressed

Reviewer #2: All comments have been addressed

2. Is the manuscript technically sound, and do the data support the conclusions?

Reviewer #1: Yes

Reviewer #2: Yes

3. Has the statistical analysis been performed appropriately and rigorously? 

Reviewer #1: Yes

Reviewer #2: Yes

4. Have the authors made all data underlying the findings in their manuscript fully available?

Reviewer #1: Yes

Reviewer #2: Yes

5. Is the manuscript presented in an intelligible fashion and written in standard English?

Reviewer #1: Yes

Reviewer #2: Yes

6. Review Comments to the Author

Reviewer #1: (No Response)

Reviewer #2: Thank you, all my recommendations have been noted.

For all my questions - hypothesis, literature, and others, all changes are noted on the text

7. PLOS authors have the option to publish the peer review history of their article (what does this mean? ). If published, this will include your full peer review and any attached files.

**Do you want your identity to be public for this peer review?** For information about this choice, including consent withdrawal, please see our Privacy Policy .

Reviewer #1: No

Reviewer #2: **Yes: ** Mariia V. Matveeva

---

## [Editor Report · Acceptance letter]

PONE-D-25-00972R1

PLOS ONE

Dear Dr. Joakimsen,

I'm pleased to inform you that your manuscript has been deemed suitable for publication in PLOS ONE. Congratulations! Your manuscript is now being handed over to our production team.

Kind regards,

on behalf of

Prof. Inge Roggen

Academic Editor

PLOS ONE